# Japanese encephalitis virus infection in non-encephalitic acute febrile illness patients

**Chairin Nisa Ma'roef[1]☯, Rama Dhenni[1]☯, Dewi Megawati[2], Araniy Fadhilah📷[1], Anton Lucanus📷[3], I Made Artika📷[1,4], Sri Masyeni[2], Asri Lestarini[2], Kartika Sari[2], Ketut Suryana[5], Frilasita A. Yudhaputri[1], Ungke Anton Jaya[1,6], R. Tedjo Sasmono[7], Jeremy P. Ledermann[8], Ann M. Powers[8], Khin Saw Aye Myint📷[1]***

**1** Emerging Virus Research Unit, Eijkman Institute for Molecular Biology, Jakarta, Indonesia, **2** Faculty of Medicine and Health Sciences, Warmadewa University, Denpasar, Bali, Indonesia, **3** School of Anatomy, Physiology and Human Biology, University of Western Australia, Perth, Australia, **4** Department of Biochemistry, Faculty of Mathematics and Natural Sciences, Bogor Agricultural University, Bogor, Indonesia, **5** Wangaya General Hospital, Denpasar, Bali, Indonesia, **6** Eijkman-Oxford Clinical Research Unit, Eijkman Institute for Molecular Biology Jakarta, Indonesia, **7** Dengue Research Unit, Eijkman Institute for Molecular Biology Jakarta, Indonesia, **8** Division of Vector-Borne Diseases, Centers for Disease Control and Prevention, Fort Collins, Colorado, United States of America

☯ These authors contributed equally to this work.
* khinsawying@hotmail.com, myintk@eijkman.go.id

**Data Availability Statement:** All relevant data are within the manuscript.

**Funding:** This work was supported by the Ministry of Research and Technology / National Agency for

## Abstract

Although Japanese encephalitis virus (JEV) is considered endemic in Indonesia, there are only limited reports of JEV infection from a small number of geographic areas within the country with the majority of these being neuroinvasive disease cases. Here, we report cases of JEV infection in non-encephalitic acute febrile illness patients from Bali, Indonesia. Paired admission (S1) and discharge (S2) serum specimens from 144 acute febrile illness patients (without evidence of acute dengue virus infection) were retrospectively tested for anti-JEV IgM antibody and confirmed by plaque reduction neutralization test (PRNT) for JEV infection. Twenty-six (18.1%) patients were anti-JEV IgM-positive or equivocal in their S2 specimens, of which 5 (3.5%) and 8 (5.6%) patients met the criteria for confirmed and probable JEV infection, respectively, based on PRNT results. Notably, these non-encephalitic JE cases were less likely to have thrombocytopenia, leukopenia, and lower hematocrit compared with confirmed dengue cases of the same cohort. These findings highlight the need to consider JEV in the diagnostic algorithm for acute febrile illnesses in endemic areas and suggest that JEV as a cause of non-encephalitic disease has likely been underestimated in Indonesia.

## Author summary

Japanese encephalitis virus (JEV) is an important cause of central nervous system (CNS) infections in Asia and is considered endemic in Indonesia. However, reports of JEV infection in non-encephalitic disease cases are lacking because diagnosis is difficult to confirm and JEV is rarely considered as a cause of non-encephalitic disease. Here, with robust serological testing, we identified cases of JEV infection in patients presenting at a regency

Research and Innovation, Republic of Indonesia
and the U.S. Centers for Disease Control and
Prevention. KSAM was awarded a grant
(5U18CK000443-05-00). The funders had no role
in study design, data collection and analysis,
decision to publish, or preparation of the
manuscript.

**Competing interests:** The authors have declared
that no competing interests exist.

hospital in Bali with fever but without symptoms of CNS infection. This finding supports
the need to include JEV in routine clinical diagnostic algorithms for patients with fever in
endemic areas.

## Introduction

Japanese encephalitis (JE) is a vector-borne disease caused by JE virus (JEV), a single-stranded
RNA flavivirus that is transmitted through a zoonotic cycle between mosquitoes, pigs and
water birds, with humans as dead-end hosts. JEV is the primary vaccine-preventable cause of
encephalitis and the major cause of viral encephalitis in Asia [1]. Clinical outcomes of symp-
tomatic JEV infection may vary from a mild non-specific febrile illness to a severe form of neu-
roinvasive disease carrying a high mortality rate (20–30%). However, most human infections
with JEV are asymptomatic, with only 1% of JEV-infected patients proceeding to develop
symptomatic clinical disease [1].

JEV is considered endemic throughout Indonesia, as suggested by early serological studies
in animals and humans [2–9], as well as from hospital-based surveillance for acute encephalitis
syndrome (AES) [10,11]. Cases of JEV infection in international tourists who have traveled to
Bali, Indonesia have also been reported [12–15]. The presence of the virus in Indonesia has
been confirmed by virus isolation from local mosquitos and pigs [16–20]. However, isolates
from confirmed human cases have yet to be reported.

Studies from Indonesia have reported that death occurred in 10–16% of laboratory-con-
firmed JE cases while 31–37% of the survivors had neurological sequelae at hospital discharge
[10,11]. Long-term assessment of Indonesian children with JE disease showed that half of the
children were either dead or left with serious disability [21]. While the AES cases are well
described, there is a limited description of symptomatic JEV infections without encephalitis. A
study from Thailand reported that 14% of adults with acute undifferentiated fevers but without
neurologic deficits were serologically diagnosed as JEV-infection [22]. It is likely that JEV is
also an under-recognized cause of fever in Indonesia. Here, with robust serological testing we
report cases of JEV infection in non-encephalitic acute febrile illness patients from Bali,
Indonesia.

## Materials and methods

### Ethics statement

This study was approved by the Medical Research Ethics Committee of Faculty of Medicine,
Udayana University, Bali (ethical approval no. 98/UN.14.2/Litbang/2015 and 1452/UN.14.2/
Litbang/2015) and the Eijkman Institute Research Ethics Commission (ethical approval no.
66). Written informed consent for participating in the study was obtained from all patients or
the parents/guardians.

### Study site, patient recruitment, and sample collection

Archived patient samples from a cross-sectional prospective study of dengue and acute febrile
illness conducted in Wangaya General Hospital (Rumah Sakit Umum Daerah Wangaya) were
analyzed. Wangaya General Hospital is located in Denpasar municipality of Bali province.
This facility includes emergency room and intensive care units as well as inpatient wards and
outpatient polyclinics for a wide variety of diseases. The hospital has 200 beds, 170,000 annual
visits, and serves a population of 880,000. Inpatient admissions come from both emergency

room and outpatient polyclinics. The leading diagnosis of patients admitted to emergency room, inpatient, and outpatient care were unspecified fever, dengue hemorrhagic fever, and diabetes mellitus, respectively. The study was conducted from March to May 2015 and September 2015 to June 2016 to study dengue and other acute febrile illnesses with the details of enrollment criteria and dengue data already reported [23]. In brief, patients presenting at the hospital with acute febrile illness (fever $\geq$38˚C with onset $\leq$7 days) but without history of chronic illnesses, human immunodeficiency syndrome, cardiac disease, sepsis, local infections (e.g. cellulitis, abscess), or gastrointestinal and respiratory symptoms were enrolled after informed consent was signed. Patients were recruited by the clinical staff by routine clinical assessment, history taking, physical examination, and laboratory tests on enrollment and/or after initial investigation. Admission blood samples (S1) were collected during patient admission while discharge blood samples (S2) were collected whenever patients were discharged from the hospital. Demographic data of the patients and clinical information were collected at the initial admission and before discharge from the hospital.

### DENV NS1 antigen detection and RT-PCR assays

DENV infection was excluded by testing of the S1 sample for DENV NS1 antigen via the SD Bioline NS1 rapid test (Alere, Australia) and DENV RNA using the Simplexa Dengue Real-time RT-PCR Kit (Diasorin, Italy) or pan-flavivirus RT-PCR as described previously [23,24]. Patients who were confirmed to have acute DENV infection were excluded from the study. Furthermore, pan-alphavirus RT-PCR was also performed to exclude chikungunya virus (CHIKV) infection as previously described [24].

### Anti-JEV and anti-DENV IgM ELISA

The presence of anti-JEV IgM in S2 specimens was detected using JEV IgM antibody capture ELISA (JEV MAC-ELISA), developed by the U.S. Centers for Disease Control and Prevention (CDC) as previously described [25]. Ratios of test serum sample optical density to negative control values (P/N) at 450 nm were calculated. Any sample with a P/N >3 was considered positive while P/N values between 2–3 were considered equivocal. Positive and equivocal S2 specimens were further re-tested paired with corresponding S1 specimens to look for seroconversion. The presence of anti-DENV IgM was also tested in both S1 and S2 specimens by using the DENV MAC-ELISA, developed by the Armed Forces Research Institute of Medical Sciences (AFRIMS), Thailand. Serum samples with DENV MAC ELISA binding index results $\geq$40 U were considered positive [26].

### Plaque reduction neutralization test (PRNT)

All specimens positive or equivocal by JEV MAC-ELISA were tested for the presence of neutralizing antibodies against JEV and DENV by PRNT. PRNT was performed with JEV strain Nakayama, DENV-1 strain PUO-359, DENV-2 strain PUO-218, DENV-3 strain PaH881/88, and DENV-4 strain 1228 as previously described [27]. Briefly, 2-fold serial dilutions of serum were mixed with each challenge virus and incubated at 37˚C for 1 hour. The antibody-virus mixture was then inoculated onto baby hamster kidney (BHK-21) cell monolayers for 5 days before plaques were counted. The neutralization titer was expressed as the inverse of the maximum serum dilution yielding a $\geq$90% reduction in plaque number (PRNT$_{90}$).

The diagnosis was classified as either confirmed JEV, probable JEV, DENV, or flavivirus infection (Table 1, adapted from [28]). Confirmed JEV infection was defined as those who were anti-JEV IgM-positive/equivocal in the S2 and/or S1 specimen with $\geq$4-fold increased anti-JEV PRNT$_{90}$ titer in S2 from S1 and an anti-JEV PRNT$_{90}$ titer $\geq$4-fold higher than any

Table 1. Diagnostic interpretation of the serology testing results.

| Diagnosis | Criteria |
|---|---|
| Confirmed JEV | Anti-JEV IgM-positive/equivocal in the S2 and/or S1, and |
| | Anti-JEV $PRNT_{90}$ titer in S2 $\geq$4-fold higher from S1, and |
| | Anti-JEV $PRNT_{90}$ titer in S2 $\geq$4-fold higher than any DENV titer |
| Probable JEV | Anti-JEV IgM-positive/equivocal in the S2 and/or S1, and |
| | Anti-JEV $PRNT_{90}$ titer in S2 higher from S1, but |
| | Anti-JEV $PRNT_{90}$ titer in S2 <4-fold than any anti-DENV titer, and |
| | Anti-DENV IgM-negative in the S1 and S2 |
| DENV | Anti-JEV IgM-positive/equivocal in the S2 and/or S1, but |
| | Anti-DENV $PRNT_{90}$ titer in S2 higher from S1, and |
| | Anti-DENV $PRNT_{90}$ titer in S2 $\geq$4-fold higher than JEV titer, and |
| | Anti-DENV IgM-positive in the S1 and/or S2 |
| Flavivirus | Anti-JEV IgM-positive/equivocal in the S2 and/or S1, but |
| | Anti-JEV $PRNT_{90}$ titer in S2 <4-fold than any anti-DENV titer, and |
| | Anti-DENV IgM-positive in the S1 and/or S2 |

S1, admission serum sample; S2, discharge serum sample; PRNT, plaque reduction neutralization test.

anti-DENV titer in S2. Probable JEV infection was defined as those who were anti-JEV IgM-positive/equivocal in the S2 and/or S1 specimen with increased anti-JEV $PRNT_{90}$ titer in S2 from S1 specimen but did not have an anti-JEV $PRNT_{90}$ titer $\geq$4-fold higher than any anti-DENV titer in S2 specimen and were anti-DENV IgM-negative in both S1 and S2 specimen. DENV infection was defined as patients who were anti-JEV IgM-positive/equivocal in the S2 and/or S1 specimen but with increased anti-DENV $PRNT_{90}$ titer in S2 from S1 specimen alongside an anti-DENV $PRNT_{90}$ titer $\geq$4-fold higher than anti-JEV titer in S2 specimen and anti-DENV IgM-positive in the S1 and/or S2 specimen. Finally, flavivirus infection was defined as those who were anti-JEV IgM-positive/equivocal in the S2 and/or S1 specimen but anti-JEV $PRNT_{90}$ titer in S2 <4-fold than any anti-DENV titer and with anti-DENV IgM-positive in the S1 and/or S2 specimen. The designation "probable JEV" was used because of the extensive cross-reactivity in secondary flavivirus infections; i.e. neutralizing antibody titers may be higher against a previous flavivirus infection rather than the most recent heterologous flavivirus infection as shown in other studies [29–33].

## Virus isolation

Virus isolation was attempted by inoculating patient S1 serum onto African green monkey kidney Vero cells as previously described [24]. Cells were observed daily for cytopathic effects for up to 10 days.

## Statistical analysis

Statistical analysis was performed using OpenEpi v3.01 and GraphPad Prism v8. Quantitative data differences between groups were compared by unpaired Student t test for normally distributed data (based on D'Agostino-Pearson normality test) or by Mann-Whitney test for non-Gaussian distributed data. Categorical data were compared using Mantel-Haenszel chi-square test when all expected numbers are at least 1 or otherwise by using Fisher's exact test. P-values less than 0.05 were considered statistically significant.

## Results

During the study period, 3,677 patients with suspected dengue or acute febrile illness attended Wangaya Hospital and a total of 703 patients were enrolled in the study (Fig 1). Of these, 321 patients had both admission (S1) and discharge (S2) paired serum specimens available and were included in this study analysis. One hundred forty-four patients showed no evidence of acute DENV infection (by detection of DENV NS1 Ag and/or RNA) nor CHIKV infection; these were tested for the presence of anti-JEV IgM. Of 144 patients, 26 (18.1%) were anti-JEV IgM-positive or equivocal in their S2 specimens (Fig 1). All 26 were further tested for the presence of anti-DENV IgM and DENV/JEV neutralizing antibodies by PRNT in paired specimens, from which 5 (3.5%) and 8 (5.6%) met the criteria for confirmed JEV and probable JEV infection, respectively. The other 5 (3.5%) and 8 (5.6%) patients were classified as having DENV and flavivirus infection, respectively (Table 2). Cell culture and pan-flavivirus RT-PCR attempted from the S1 specimens did not produce any JEV positive results.

Two patients (WGY599 and 623) with confirmed JEV infection diagnosis had no detectable or low neutralizing antibodies to JEV or any DENV in their S1 specimen, suggesting that JEV was the first flavivirus that they encountered (primary flavivirus/JEV). These two patients had IgM-positive results to both JEV and DENV, however a $\geq$4-fold rise of anti-JEV $PRNT_{90}$ titer and a $\geq$4-fold difference between anti-JEV $PRNT_{90}$ titer and anti-DENVs titers were observed in their S2 specimen, indicating a low level of cross-reactivity in the cases of primary JEV/flavivirus infection.

Most of the patients with probable JEV diagnosis had neutralizing antibody response to DENV in their S1 sample, suggesting previous exposure to DENV. Although most of these probable JEV-infected patients had a higher neutralizing antibody titer to DENV compared to JEV in their S2 specimen, we believe that these patients were infected with JEV since all had no detectable DENV IgM in both S1 and S2 specimens with positive or equivocal JEV IgM in the S2 specimen. The phenomenon of higher serologic reactivity to the previous flavivirus infection than to the current infection is known as "original antigenic sin" and has been documented in a number of studies in human and animal models [29–33]. The mechanism of this phenomenon is not completely understood but is likely attributed to a preferential expansion of memory B cell clones generated from previous flavivirus infection that cross-react and recognize the current infecting heterologous flavivirus [34]. In addition, the absence of DENV RNA detectable in S1 specimens by highly sensitive RT-PCR provides further evidence of JE diagnosis as opposed to dengue.

Four out of five (80%) patients diagnosed as DENV infection had high PRNT titer ($\geq$1,280) to one or more DENV serotypes with detectable anti-DENV IgM in the S2 specimen. Similar to probable JEV-infected patients, neutralizing antibody against DENV was detectable in their S1 which suggest secondary DENV infection. This was also observed in patients with flavivirus infection diagnosis. However, in the flavivirus group, serological data is not sufficient to distinguish the likely infecting flavivirus.

The clinical characteristics of JEV-infected patients are provided in Table 3 along with 177 confirmed DENV-infected patients identified by DENV NS1 antigen detection and/or RT-PCR from the same study cohort (Fig 1). The age of the JE patients ranged from 4 to 72 years old (median 23 years old) and eight of 13 patients were male (62%) (Table 3). The most notable symptoms were malaise, nausea, and loss of appetite, which was observed in 85%, 69%, and 54% of the patients, respectively. Hematological investigations showed thrombocytopenia and leukopenia in nine (69%) and seven (54%) patients, respectively. Mean nadir thrombocyte count was $107 \pm 74.2 \times 10^3/\mu l$ (median $94 \times 10^3/\mu l$) while mean nadir leukocyte count was $4 \pm 1 \times 10^3/\mu l$ (median $4 \times 10^3/\mu l$) among all 13 patients. No neurological manifestations

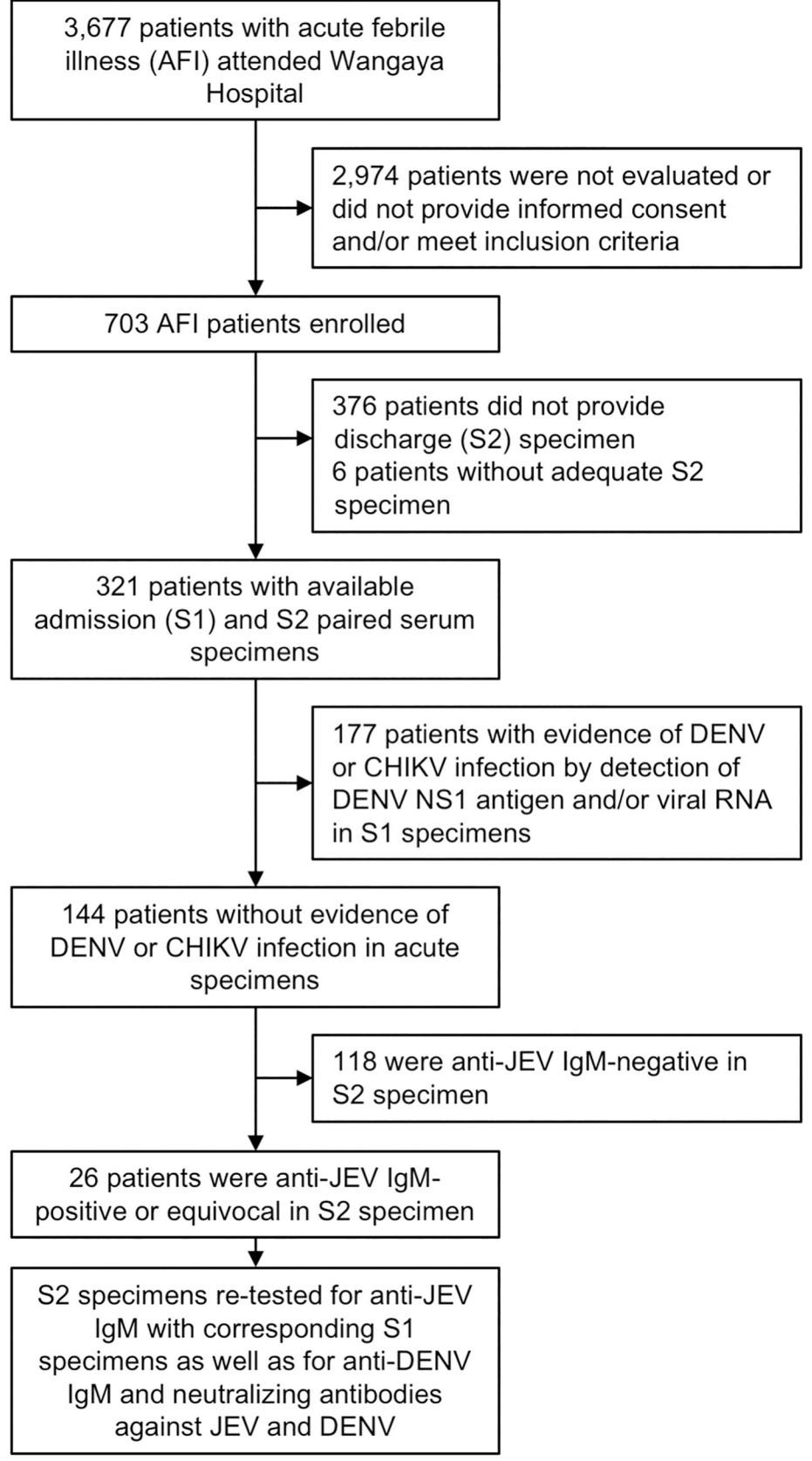

**Fig 1. Patient enrollment, specimens collected, and molecular and serological test performed.** DENV NS1 antigen was detected by using SD Bioline NS1 rapid test while DENV RNA was detected using Simplexa Dengue Real-time RT-PCR Kit or pan-flavivirus RT-PCR. CHIKV RNA was detected using pan-alphavirus RT-PCR. Anti-JEV and anti-DENV IgM was detected by ELISA, while neutralizing antibodies against JEV and DENV were detected by plaque reduction neutralization test (PRNT).

**Table 2. Serology testing results from non-encephalitic acute febrile illness patients with confirmed JEV, probable JEV, DENV, and flavivirus infection.**

| No. | Patient sample ID | Age (years) | Days after onset | IgM | | PRNT$_{90}$ titer | | | | |
|---|---|---|---|---|---|---|---|---|---|---|
| | | | | JEV[a] | DENV[b] | JEV | DENV-1 | DENV-2 | DENV-3 | DENV-4 |
| JEV (confirmed) | | | | | | | | | | |
| 1 | WGY599 S1 | 20 | 3 | − | − | 40 | <10 | <10 | <10 | <10 |
| | WGY599 S2 | | 6 | + | + | 1,280 | 20 | 80 | 20 | 20 |
| 2 | WGY623 S1 | 32 | 3 | − | − | <10 | <10 | <10 | <10 | <10 |
| | WGY623 S2 | | 7 | + | + | 640 | 40 | 40 | 80 | 80 |
| 3 | WGY656 S1 | 23 | 2 | EQU | − | 20 | <10 | 80 | 10 | 20 |
| | WGY656 S2 | | 3 | + | − | >320 | 40 | 40 | 40 | NA |
| 4 | WGY726 S1 | 32 | 7 | + | + | 80 | 10 | 40 | 80 | 10 |
| | WGY726 S2 | | 10 | + | + | 320 | 10 | 40 | 20 | 20 |
| 5 | WGY733 S1 | 15 | 4 | − | − | 80 | <10 | 40 | 80 | 20 |
| | WGY733 S2 | | 7 | + | + | 320 | 10 | 40 | 80 | 20 |
| JEV (probable) | | | | | | | | | | |
| 6 | WGY067 S1 | 22 | 4 | QNS | − | 40 | 160 | 80 | 80 | 80 |
| | WGY067 S2 | | 7 | + | − | 80 | >2,560 | 160 | 160 | 160 |
| 7 | WGY143 S1 | 4 | 7 | QNS | − | 40 | 640 | 80 | 80 | 20 |
| | WGY143 S2 | | 12 | EQU | − | 160 | >2,560 | 160 | 160 | 80 |
| 8 | WGY317 S1 | 47 | 5 | + | − | 40 | <10 | 80 | 10 | 40 |
| | WGY317 S2 | | 8 | + | − | 80 | <10 | 160 | 10 | 80 |
| 9 | WGY629 S1 | 19 | 5 | + | − | 160 | 20 | 160 | 80 | 160 |
| | WGY629 S2 | | 7 | + | − | 320 | 320 | 1,280 | 160 | 320 |
| 10 | WGY647 S1 | 27 | 5 | + | − | 80 | 160 | 80 | 40 | 80 |
| | WGY647 S2 | | 6 | + | − | 160 | 1,280 | 20 | 160 | 80 |
| 11 | WGY658 S1 | 30 | 3 | EQU | − | <10 | 40 | <10 | <10 | 10 |
| | WGY658 S2 | | 5 | EQU | − | 80 | 80 | <10 | <10 | <10 |
| 12 | WGY667 S1 | 6 | 5 | EQU | − | 20 | <10 | 20 | 10 | 40 |
| | WGY667 S2 | | 7 | + | − | 160 | 20 | 160 | 160 | 160 |
| 13 | WGY713 S1 | 72 | 3 | EQU | − | 10 | NA | 40 | NA | NA |
| | WGY713 S2 | | 8 | EQU | − | 80 | 40 | 320 | 40 | 40 |
| DENV | | | | | | | | | | |
| 14 | WGY065 S1 | 33 | 5 | QNS | − | 320 | 160 | 160 | 320 | 160 |
| | WGY065 S2 | | 8 | + | + | 320 | >2,560 | >2,560 | >2,560 | 160 |
| 15 | WGY071 S1 | 16 | 4 | QNS | − | 20 | 10 | 40 | 160 | 20 |
| | WGY071 S2 | | 7 | + | + | 20 | 80 | 160 | >2,560 | 80 |
| 16 | WGY321 S1 | 24 | 5 | − | − | 160 | 160 | 40 | 80 | 20 |
| | WGY321 S2 | | 7 | + | + | 160 | 1280 | 160 | 160 | 80 |
| 17 | WGY724 S1 | 28 | 4 | EQU | − | <10 | 40 | 10 | 20 | 10 |
| | WGY724 S2 | | 7 | EQU | + | 20 | 80 | 160 | 80 | 80 |
| 18 | WGY727 S1 | 19 | 5 | − | + | 20 | 40 | 40 | 40 | 40 |
| | WGY727 S2 | | 6 | EQU | + | 160 | >2,560 | 80 | 80 | 160 |

*(Continued)*

**Table 2.** (Continued)

| No. | Patient sample ID | Age (years) | Days after onset | IgM | | PRNT$_{90}$ titer | | | | |
|---|---|---|---|---|---|---|---|---|---|---|
| | | | | JEV[a] | DENV[b] | JEV | DENV-1 | DENV-2 | DENV-3 | DENV-4 |
| Flavivirus | | | | | | | | | | |
| 19 | WGY089 S1 | 33 | 5 | QNS | + | 40 | 20 | 80 | 80 | 40 |
| | WGY089 S2 | | 8 | + | + | 80 | 20 | 80 | 160 | 40 |
| 20 | WGY174 S1 | 3 | 4 | + | + | 40 | 10 | 80 | 80 | 160 |
| | WGY174 S2 | | 5 | + | + | 320 | 10 | 320 | 80 | 320 |
| 21 | WGY178 S1 | 20 | 3 | + | − | 80 | 40 | 80 | <10 | <10 |
| | WGY178 S2 | | 5 | + | + | 160 | 40 | 320 | 20 | 20 |
| 22 | WGY516 S1 | 15 | 3 | QNS | + | <10 | <10 | <10 | 40 | <10 |
| | WGY516 S2 | | 5 | EQU | + | 20 | <10 | <10 | 40 | <10 |
| 23 | WGY575 S1 | 26 | 4 | EQU | − | 20 | 20 | 40 | 20 | 10 |
| | WGY575 S2 | | 5 | + | + | 160 | 40 | 80 | 160 | 40 |
| 24 | WGY674 S1 | 21 | 5 | − | − | 40 | <10 | <10 | <10 | <10 |
| | WGY674 S2 | | 7 | + | + | 40 | 10 | 20 | <10 | 20 |
| 25 | WGY684 S1 | 16 | 5 | + | + | 80 | <10 | 10 | 40 | 10 |
| | WGY684 S2 | | 6 | + | + | 80 | 10 | 20 | 40 | 20 |
| 26 | WGY708 S1 | 31 | 3 | + | − | NA | NA | NA | NA | NA |
| | WGY708 S2 | | 6 | + | + | 40 | 40 | 40 | 20 | 40 |

[a]Ratios of samples optical densities divided by negative control were calculated; <2, negative (−); 2–3 equivocal (EQU); >3 positive (+).

[b]Calculated ELISA binding index results ≥40 U were considered positive (+), while <40 U were considered negative (−).

S1, admission serum sample; S2, discharge serum sample; QNS, quantity not sufficient.

or rashes were reported. Notably, there were no statistically significant differences found in clinical characteristics between confirmed and probable JE cases except for lowest hemoglobin level (14.5 ± 1.3 vs 12.6 ± 1.3 g/dl, $P = 0.0319$).

Interestingly when compared with the dengue patients, JE patients were less likely to have leukopenia (54% vs 82%, $P = 0.018$), thrombocytopenia (69% vs 92%, $P = 0.007$), or headache (31% vs 59%, $P = 0.045$). JE patients were also found to have a higher hematocrit than dengue patients (median 47% vs 43%, $P = 0.030$) (Table 3). Furthermore, although our study plans did not include follow up assessment to determine long term outcome of the patients after discharge, all JE patients had resolved symptoms upon hospital discharge.

It is not clear from our data if there was any temporal distribution pattern of the JEV infection since our study was not conducted throughout the whole year and the limited number of identified JE cases prevented making such an analysis. However, out of the 13 JE cases, six cases occurred in May, two cases in July, and each one case in March, April, June, September, and October.

## Discussion

JEV was not previously considered a significant public health problem in Indonesia until nationwide studies in the early 2000s (based on syndromic surveillance and serologic assays) suggested nationwide JEV endemicity [9–11]. Although there are a number of laboratory tests to diagnose JEV infection, virus detection assays are not useful for diagnostic purposes due to low-level, transient viremia, making anti-JEV IgM ELISA the WHO recommended method for JEV diagnosis and surveillance [35]. However, cross-reactive IgM antibodies have been detected in about 10% of DENV and JEV cases [36,37]. Therefore, a conservative case

**Table 3. Characteristics of patients with JEV and DENV infection identified in this study.**

| Patient characteristics | | JEV (n = 13)[a] | DENV (n = 177)[b] |
|---|---|---|---|
| Demographic | | | |
| | Sex, male | 8 (62%) | 84 (48%) |
| | Age | 23 (4–72) | 24 (1–75) |
| Duration of illness before admission, days | | 4 (2–7) | 4 (2–7) |
| Hematology | | | |
| | Lowest hemoglobin, g/dl | 13 (10–16) | 13 (9–17) |
| | Lowest leukocyte counts, $10^3$/μl | 4 (2–5)[c] | 3 (1–8) |
| | Leukopenia (<4,000/μl) | 7 (54%)[d] | 144 (82%) |
| | Lowest thrombocyte counts, $10^3$/μl | 94 (24–258) | 53 (0.03–508) |
| | Thrombocytopenia (<150,000/μl) | 9 (69%)[e] | 163 (92%) |
| | Highest hematocrit, % | 47 (33–89)[f] | 43 (6–164) |
| Clinical symptoms | | | |
| | Malaise | 11 (85%) | 153 (86%) |
| | Nausea | 9 (69%) | 143 (81%) |
| | Loss of appetite | 7 (54%) | 125 (71%) |
| | Myalgia | 5 (39%) | 87 (49%) |
| | Headache | 4 (31%)[g] | 105 (59%) |
| | Hemorrhage | 3 (23%) | 28 (16%) |
| | Arthralgia | 2 (15%) | 71 (40%) |
| | Vomiting | 2 (15%) | 72 (41%) |
| | Cough | 1 (8%) | 15 (9%) |
| | Retro-orbital pain | 1 (8%) | 38 (22%) |
| | Rash | 0 (0%) | 11 (6%) |
| | Diarrhea | 0 (0%) | 4 (2%) |
| | Runny nose | 0 (0%) | 4 (2%) |
| | Neck stiffness | 0 (0%) | 1 (1%) |
| | Seizure | 0 (0%) | 0 (0%) |
| | Abdominal pain | 0 (0%) | 25 (14%) |
| | Altered mental status | 0 (0%) | 0 (0%) |
| | Paralysis | 0 (0%) | 0 (0%) |
| | Glasgow Coma Scale <15 | 0 (0%) | 0 (0%) |

Data are presented as number of patients (%) or median (range).

[a]Includes confirmed JEV (n = 5) and probable JEV (n = 8) infection.

[b]Includes confirmed DENV-infected patients identified by detection of DENV NS1 antigen detection and/or RT-PCR as depicted in Fig 1.

[c]Statistically significant compared with DENV ($P$ = 0.002).

[d]Statistically significant compared with DENV ($P$ = 0.018).

[e]Statistically significant compared with DENV ($P$ = 0.007).

[f]Statistically significant compared with DENV ($P$ = 0.003).

[g]Statistically significant compared with DENV ($P$ = 0.045).

definition was used here to define JEV infection based on IgM ELISA followed by confirmation with PRNT in both admission and discharge serum samples. This study confirmed JEV as a cause of non-encephalitic acute febrile illness in Bali, where both JEV and DENV co-circulate.

In this study population, confirmed and probable JEV infection were identified in 9% (13 out of 144) cases. From the thirteen JE patients diagnosed in this study, eleven were adults while only two were children. Previous studies showed that more than 80% of Indonesian

children have experienced DENV infection at least once before the age of ten [38], which likely explains the low prevalence of cases with no detectable or low neutralizing antibodies to JEV or any DENV in their S1 specimen (i.e. primary JEV/flavivirus infection) in our study. Further, the presence of pre-existing DENV antibodies in JEV-infected patients has recently been associated with better patient outcomes [39]. Hence, the absence of severe or encephalitic disease in these subjects could be partly attributed to pre-existing DENV immunity.

Thrombocytopenia, prevalent in DENV-infected patients identified in this study, was also observed in 69% of the febrile JE cases, similar to the non-encephalitic JEV infections from Thailand [22]. Malaise, nausea, loss of appetite, myalgia, and headache were the major symptoms reported in the JEV cases here, similar to those reported previously [22]. However, these symptoms were also present in DENV-infected patients at similar frequency except for headache which was less observed in JE cases. While this study suggests that thrombocytopenia, leukopenia, and lower hematocrit were less likely to be found in non-encephalitic JE compared with dengue cases, further studies are needed to confirm these findings.

JEV is routinely included in the diagnostic algorithm of AES in endemic areas of Indonesia. However, reports of JEV as the cause of non-encephalitic illness by using a virus specific PRNT confirmatory assay are lacking in Indonesia. The use of PRNT in this study was vital in confirming JEV infection especially in cases where anti-JEV and -DENV IgM were both detected as exemplified in patients WGY599, 623, 726, and 733. Unfortunately, there is limited laboratory capacity in Indonesia to perform PRNT or detect flaviviruses other than DENV. The role of other vector-borne viruses, including JEV, as causes of febrile illness or encephalitis has therefore likely been underestimated. As such, JEV remains an important public health concern in Indonesia and the transmission of JEV warrants further investigation.

This study is limited by the number of non-encephalitic JE cases identified which does not allow for a sound stratified analysis of the results particularly regarding the clinical features. Furthermore, the higher prevalence of adults over children identified in the study might be due to lack of appropriate population denominator data. The incidence of non-encephalitic JE might potentially be higher in children if the data from this study were adjusted by age stratified population denominator (i.e. the number of susceptible children during the study period).

In summary, this work demonstrates JEV infection in non-encephalitic acute febrile illness patients identified using robust serological assays. Although JEV vaccination has recently been introduced in Bali [40] with reported coverage of 94% in 2018 [41], it has not been widely implemented throughout Indonesia. Hence, further JEV surveillance is required to fully reveal the epidemiology of JE disease in humans. This report on JEV as the cause of acute febrile illness in Bali is fundamental to characterizing JE epidemiology, identifying high-risk areas, and documenting the impact of prevention measures in Indonesia.

## Acknowledgments

The authors would like to thank the patients, physicians, and the management of Wangaya General Hospital, Denpasar, Bali for their support during the acute febrile illness study. We thank Made Satya Dharmayanti of the Biomolecular Laboratory of Warmadewa University, for her help in diagnostic testing and specimen archiving. We are indebted to the Department of Virology, Armed Forces Research Institute of Medical Sciences, Bangkok, Thailand, for their support on the DENV MAC-ELISA kit used in this study. This work was supported by the Ministry of Research and Technology / National Agency for Research and Innovation, Republic of Indonesia, and the U.S. Centers for Disease Control and Prevention. The contents and conclusions of this report are those of the authors and do not necessarily represent the official position of the U.S. Centers for Disease Control and Prevention.

## Author Contributions

**Conceptualization:** Dewi Megawati, Ann M. Powers, Khin Saw Aye Myint.

**Data curation:** Rama Dhenni, Dewi Megawati, Sri Masyeni, Asri Lestarini, Kartika Sari, Ungke Anton Jaya.

**Formal analysis:** Chairin Nisa Ma'roef, Rama Dhenni, Araniy Fadhilah, Anton Lucanus.

**Funding acquisition:** Ann M. Powers, Khin Saw Aye Myint.

**Investigation:** Chairin Nisa Ma'roef, Rama Dhenni, Dewi Megawati, Araniy Fadhilah, Anton Lucanus, Asri Lestarini, Kartika Sari.

**Methodology:** Chairin Nisa Ma'roef, Rama Dhenni, Araniy Fadhilah, Anton Lucanus, Jeremy P. Ledermann, Ann M. Powers, Khin Saw Aye Myint.

**Project administration:** Dewi Megawati, I Made Artika, Asri Lestarini, Kartika Sari, Ketut Suryana, Frilasita A. Yudhaputri, Ungke Anton Jaya, Khin Saw Aye Myint.

**Resources:** Dewi Megawati, Sri Masyeni, Asri Lestarini, Kartika Sari, Ketut Suryana, R. Tedjo Sasmono, Ann M. Powers, Khin Saw Aye Myint.

**Supervision:** I Made Artika, Frilasita A. Yudhaputri, Ungke Anton Jaya, Jeremy P. Ledermann, Ann M. Powers, Khin Saw Aye Myint.

**Visualization:** Rama Dhenni.

**Writing – original draft:** Rama Dhenni, Khin Saw Aye Myint.

**Writing – review & editing:** Rama Dhenni, Anton Lucanus, R. Tedjo Sasmono, Jeremy P. Ledermann, Ann M. Powers, Khin Saw Aye Myint.

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
