## [Decision Letter · Decision Letter 0]

20 Apr 2020

Dear Myint,

Thank you very much for submitting your manuscript "Japanese Encephalitis Virus Infection in Non-Encephalitic Acute Febrile Illness Patients" for consideration at PLOS Neglected Tropical Diseases. As with all papers reviewed by the journal, your manuscript was reviewed by members of the editorial board and by several independent reviewers. In light of the reviews (below this email), we would like to invite the resubmission of a significantly-revised version that takes into account the reviewers' comments. 

AE comments:

1) Please be sure to respond to the reviewer comments sent as separate files.

2) Please provide a brief description of the collection of clinical and clinical laboratory data (Table 2), e.g., when and how frequently this was collected.

3) Please comment on whether the clinical characteristics were similar in the cases defined as confirmed and probably JE (Table 2).

We cannot make any decision about publication until we have seen the revised manuscript and your response to the reviewers' comments. Your revised manuscript is also likely to be sent to reviewers for further evaluation.

Sincerely,

Alan L Rothman, MD

Associate Editor

Scott Weaver

Deputy Editor

AE comments:

1) Please be sure to respond to the reviewer comments sent as separate files.

2) Please provide a brief description of the collection of clinical and clinical laboratory data (Table 2), e.g., when and how frequently this was collected.

3) Please comment on whether the clinical characteristics were similar in the cases defined as confirmed and probably JE (Table 2).

Reviewer's Responses to Questions

**Key Review Criteria Required for Acceptance?**

**Methods**

-Are the objectives of the study clearly articulated with a clear testable hypothesis stated?

-Is the study design appropriate to address the stated objectives?

-Is the population clearly described and appropriate for the hypothesis being tested?

-Is the sample size sufficient to ensure adequate power to address the hypothesis being tested?

-Were correct statistical analysis used to support conclusions?

-Are there concerns about ethical or regulatory requirements being met?

Reviewer #1: The study had clearly stated objectives and design for an observation in a hospital population. I believe the study reports on a sufficient number of cases to support the findings but the nature of the study does not requirer statistical analyses. My only comment on design is an editorial one. Use of RT-PCR for dengue analysis of S1 is described under "Study site, patient recruitment and sample collection". This is fine, but use of PCR should appear under Methods in the same way that Elisa, PRNT etc. do. Either the description of Rt-PCR should be moved there under a heading of Rt-PCR or there should be a note in the methods under an RT-PCR heading that refers to the description in the Study site section. I think the former is preferable. I have no concerns about ethical conduct of the study.

Reviewer #2: The objective of the study was articulated well and the study design was appropriate for addressing the objective.

Reviewer #3: (No Response)

**Results**

-Does the analysis presented match the analysis plan?

-Are the results clearly and completely presented?

-Are the figures (Tables, Images) of sufficient quality for clarity?

Reviewer #1: I believe that the results are presented clearly and match the analysis plan. The category criteria for confirmed and probable JE or dengue are, of necessity, complex and the textual explanation of why some probable JE cases are not dengue is critical and helps one understand the small +/- data presented in the Table. I would suggest, however that the RT-PCR data for dengue be cited in the results as further evidence in support of those probable cases fitting JE more closely, because RT-PCR is highly sensitive and specific in the febrile period for dengue diagnosis. It is reported in a sentence in the manuscript that there were no positive RT-PCR for dengue or JE. That is unsurprising for JE as it is quite rare to have high enough sustained viremia to detect JE infection with PCR, but not so with dengue. The fact that PCR was negative for dengue in the Probably JE cases adds to the argument made on the basis of serologic criteria that the JE probable cases were in fact more likely to be JE. I think it helps your case if you state that clearly in this section of the results.

Reviewer #2: The results of the study were presented clearly, however, there was room for inclusion of more information.

Reviewer #3: (No Response)

**Conclusions**

-Are the conclusions supported by the data presented?

-Are the limitations of analysis clearly described?

-Do the authors discuss how these data can be helpful to advance our understanding of the topic under study?

-Is public health relevance addressed?

Reviewer #1: The conclusions are supported by the data and I believe, as noted above, reporting of RT-PCR negative dengue tests in the results would make an even stronger case. There is no real discussion of limitations other than the difficulty of flavivirus serologic cross-reactivity. The small numbers don't allow for any stratification of results and statistical analysis and that should be described as a limitation. I think the finding of a clinical, non-encephalitic state for JE, along with that described in reference 22, is very important. Most people think JE presents only as severe illness or asymptomatic infection. It is helpful to understand that there is, not surprisingly, an intermediate circumstance. My last comment is that the authors should consider that the odd adult distribution is likely due to the fact that this is numerator only study. Seeming predominance of adult cases of JE is common where only numerator data is available. The 15-20% of individuals who are not infected as children and acquire the virus later in life can be quite numerous in a population. In most circumstances where vaccine has not been broadly applied, adjusting the numerator data by age stratified population denominators shows that the incidence is much higher in children under 15 years of age. The authors should address the lack of population denominator data as another key limitation of the study.

Reviewer #2: The conclusion of the study was supported by the data presented.

Reviewer #3: (No Response)

**Editorial and Data Presentation Modifications?**

Reviewer #1: I felt that the editorial suggestions that I have are better addressed in the context of the questions posed for review in the above sections and have taken that approach. I have no more suggestions here.

Reviewer #2: (No Response)

Reviewer #3: (No Response)

**Summary and General Comments**

Reviewer #1: I think this is of high significance and am grateful to see that the study has been undertaken. I have seen such information from fever surveillance in one country where clinical symptoms were not readily available. This study makes it clear that these cases are very likely JE and that they have little symptomatology other, than the 30% of cases with headache, that would make any one think of JE encephalitis. It is very helpful to broaden the understanding of the spectrum of JE disease.

Reviewer #2: This would be a missed opportunity if some additional results were not included as mentioned in the review comments.

Reviewer #3: The draft manuscript describes a retrospective analyses of paired-serum samples collected from febrile, not encephalitic, patients in Bali, Indonesia. A significant proportion of these samples demonstrate seroconversion to JEV, perhaps underscoring the need of more robust surveillance in the region and the need for JEV vaccinations. Nonetheless, as is, the draft lacks more thorough and consistent testing to properly provide for evidence of JEV seroconversions. Before acceptance, authors should be given an opportunity to address findings in attached letter.

PLOS authors have the option to publish the peer review history of their article (what does this mean?). If published, this will include your full peer review and any attached files.

Reviewer #1: No

Reviewer #2: No

Reviewer #3: No
---

## [Editor Report · Decision Letter 1]

5 Jun 2020

Dear Myint,

We are pleased to inform you that your manuscript 'Japanese Encephalitis Virus Infection in Non-Encephalitic Acute Febrile Illness Patients' has been provisionally accepted for publication in PLOS Neglected Tropical Diseases.

Best regards,

Alan L Rothman, MD

Associate Editor

Scott Weaver

Deputy Editor

---

## [Editor Report · Acceptance letter]

24 Jun 2020

Dear Myint,

We are delighted to inform you that your manuscript, "Japanese Encephalitis Virus Infection in Non-Encephalitic Acute Febrile Illness Patients," has been formally accepted for publication in PLOS Neglected Tropical Diseases.

Best regards,

Shaden Kamhawi

co-Editor-in-Chief

Paul Brindley

co-Editor-in-Chief
